

# lncRNA–mRNA competing endogenous RNA network in IR-hepG2 cells ameliorated by APBBR decreasing ROS levels: a systematic analysis

Min Lin[1] and Zhu-Jun Mao[2]

[1] College of Basic Medicine, Zhejiang Chinese Medical University, Hangzhou, China
[2] College of Pharmaceutical Sciences, Zhejiang Chinese Medical University, Hangzhou, China

Corresponding author
Zhu-Jun Mao, maozhujun0107@zcmu.edu.cn, maozhujun0107@163.com

## ABSTRACT

**Background**. Radix Astragali (Astragalus membranaceus var. mongholicus (Bunge)) and Coptis chinensis (Coptis chinensis var. angustiloba) are two commonly prescribed traditional Chinese herbs for diabetes. Astragalus Polysaccharide (AP) and Berberine (BBR) are active ingredients of these two herbs respectively and they are scientifically proved to have immunomodulatory and anti-inflammatory effects. They are also known for their antidiabetic potential by ameliorating insulin resistance (IR). AP and BBR have shown different advantages in treating diabetes according to previous reports. However, very few studies focus on the combined activities of the two potential antidiabetic ingredients. In this study, we discovered that reactive oxygen species (ROS) accumulated in IR-hepG2 cells and APBBR can decrease ROS level in model group significantly. We conjectured that APBBR can ameliorate IR in hepG2 cells by decreasing ROS level. In order to verify this hypothesis, we obtained phenotype and transcriptome information of IR-HepG2 cells and explore the underlying mechanism of the combination of AP and BBR(APBBR) activity on the relationship between ROS change in IR at whole-transcriptome level, so as to shed new light to efficacy and application of APBBR in treating diabetes.

**Methods**. The IR cell model was established with high-level insulin intervention. Glucose content, HepG2 cell viability as well as ROS level was detected to study the effect of IR-hepG2 cell phenotype. Unbiased genome-wide RNA sequencing was used to investigate alterations in experimental groups. Then, GO and KEGG functional enrichment was performed to explore the function and pathway of target genes. Venn analysis found out the differentially expressed lncRNAs that had close relationship with IR and ROS. Finally, we screened out candidate lncRNAs and these target genes to construct interaction network of differentiated lncRNA–miRNA–mRNA by according to the principle of competitive endogenous RNA (ceRNA).

**Results**. The biochemical experiments showed that APBBR administration could improve the proliferation activity of IR-HepG2 cells and decrease ROS level in model cells. The GO and KEGG functional enrichment analyses demonstrated several mRNAs remarkably enriched in biological processes and signaling pathways related to ROS production and IR progression. Interaction network suggest that APBBR ameliorates IR in HepG2 cells by regulating the expression of multiple genes and activating relevant signaling pathway to decrease ROS level. Thus, we demonstrated that APBBR ameliorated IR in hepG2 cells via the ROS-dependent pathway.

## INTRODUCTION

Transcriptome sequencing in various organisms has revealed that abundant transcription derived from human transcriptome generates a large proportion of non-coding RNAs (ncRNAs). Increasing evidence have shown the critical regulatory potential of ncRNAs in biological processes. The ncRNAs are classified into two categories according to their length: long non-coding RNAs (lncRNAs) with a length of more than 200 nucleotides and short noncoding RNAs (18 to 200 nucleotides), respectively (*Li et al., 2019*). lncRNA accounts for 68% of ncRNAs (*Iyer et al., 2015*), but they were considered ''junk RNA'' in the genome because they could not encode proteins (*Zhang et al., 2019*). However, recent studies have shown that lncRNA exert critical roles in regulating biological processes through comprehensive mechanisms, including genetic imprinting, immune response, tumorigenesis, cellular development and metabolism (*Liyanage & Ganegoda, 2017*). LncRNAs are also involved in the regulation of glucose and lipid metabolism in the liver, and their abnormal expression may be involved in the development of type 2 diabetes mellitus (T2DM) (*Eslam, Valenti & Romeo, 2018*).

Insulin resistance (IR)refers to a physiological condition in which target tissue such as fat, liver and muscle are unable to use insulin effectively. The metabolism of substances in the body is almost all regulated by insulin (*Williams & Olefsky, 1990*). Once IR occurs, the metabolism of sugars, lipids and proteins will be disturbed. The physiological regulation of insulin is mainly carried out by combining with the insulin receptors on the cells of various tissues. Any defect in the signal transmission in the process may affect the normal physiological regulation process of insulin and thus cause the occurrence of IR. IR degree in different tissues is specific, which makes its mechanism complex.

Reactive oxygen species (ROS) is a collective term to include oxygen-containing reactive species (*Li, Jia & Trush, 2016*). It is a significant contributor to the cell and tissue dysfunction in diabetes and the progression of complex diabetic complications (*Newsholme et al., 2007*). The increased ROS formation may cause oxidative stress, which plays a significant role in the development of hepatic IR (*Liu et al., 2017*). HepG2 cells are ideal model for studying hepatic IR pathogenesis and action mechanism of hypoglycemic drugs. The number of insulin receptors on the surface of HepG2 cells decreases with high level insulin intervention (*Williams & Olefsky, 1990*). At present, the underlying molecular mechanisms and regulation of ROS change in IR-HepG2 cells remains poorly understood.

Astragalus Polysaccharide (AP) is the main effective ingredient extracted from the Chinese herb Radix Astragali (Astragalus membranaceus var. mongholicus (Bunge)), which has the potential activity of antioxidant effects (*Fu et al., 2018*). Berberine (BBR) is an isoquinoline extracted from Coptis chinensis (Coptis chinensis var. angustiloba), which has many pharmacological activities such as anti-microbial, anti-diarrheal, reducing glucose and cholesterol, anti-tumor and immunosuppression. The plant names have been

checked with http://www.theplantlist.org. Both of these two herbs have been used to treat diabetes for thousands of years in Traditional Chinese Medicine, study showed that the combination of Coptidis rhizoma, Astragali radix and Lonicera japonica(Lonicera alba L.) has a synergistic effect by increasing insulin sensitivity and ameliorating IR (*Yang, 2017*). In addition, it has been verified that AP and BBR could promote glucose up-take and improve carbohydrate metabolism respectively with different characteristics (*Wang et al., 2019*). Our previous study results showed that AP and BBR used together at a ratio of 1:1 could promote the basic secretion ability of IR-INS-1 cells model significantly, and showed a synergistic effect in diabetes treating (*Mao, Shou & Chai, 2017*). The study found has been applied for a patent, the application number: CN201810266347.9, publication/announcement number: CN108478591A. However, the cellular and molecular compatibility mechanism responsible for APBBR in attenuating IR remains unclear.

Additionally, there were few studies to identify key genes and pathways involved in ROS change in IR-HepG2 cells. Spurred by the development of deep sequencing technology, we are allowed to investigate the mechanisms of hepatic IR at the whole-transcriptome level. In the present study, we used high-throughput RNA-seq technology to explore the potential mechanism underlying hepatic IR at whole-transcriptome level by analyzing changes in the global gene expression profile in normal, IR and APBBR-treated HepG2 cells for the first time. First, differentially expressed genes (DEGs), related gene ontology (GO) and pathways were determined. Subsequently, gene act, pathway act and co-expression network were constructed to further explore the role of the ROS and IR related genes and pathways, which could shed new light to the underlying mechanisms of ROS and IR and therapeutic applications of APBBR in treating diabetes.

## MATERIALS & METHODS

### HepG2 cells cultured

HepG2 cells (Cell Bank of the Chinese Academy of Sciences, Shanghai, China) were cultured in 1640 medium (Hyclone, Beijing, China) containing 10% fetal bovine serum (FBS, Hyclone, Beijing, China) and $1\times$ streptomycin in a 37 °C 5% CO2 saturated humidity incubator. The normally cultured HepG2 cell lines in log phase were centrifuged at 100 grpm for 5 min. 20,000 cells/well were plated in a 96-well plate and incubated at 37 °C.

### Administration

Insulin (Gibco, NY, USA) was diluted to a final concentration of $10^{-6}$ mol/L according to our preliminary experiment. 200 µl of the insulin preparation was added into each well for the model group and 200 µl of complete media was added for the control group. Culture was performed in 37 ° C, 5% $CO_2$ and saturated humidity incubator. The supernatant of the corresponding medium was collected after 48 h by centrifugation at 3000 r/min for 5 min and stored at $-80$ °C for use .

### Cell groups
#### Control, Model, APBBR
To examine whether the APBBR had effect on HepG2 cells, HepG2 cells assigned to a control group(Control), and the molding cells were assigned randomly into IR model

group(Model); Model + 1 mg APBBR group (1 mg APBBR group); Model + 5 mg APBBR group (5 mg APBBR group), Model + 10 mg APBBR group (10 mg APBBR group); Model + 20 mg APBBR group (20 mg APBBR group); Model + 40 mg APBBR group (40 mg APBBR group). APBBR was administered at a 1:1 mass ratio of AP:BBR. After successful establishment of the IR model, different concentrations of APBBR were added. (The mass ratio of AP to BBR was 1:1,Astragalus Polysaccharide was provided by Lot No. C11M7Y10255, Shanghai Yuanye Biotechnology Co., Ltd.Shanghai, China. Assay ≥98%; Berberine was provided by Lot No.160528, Beijing Century Aoko Biotechnology Co., Ltd. Beijing, China. HPLC ≥98%); Control and Model group was cultured in normal medium, each well system was 2 ml, and the action time was 48 h.

## Study on the effect of IR-hepG2 cell phenotype
### Glucose content determination
The glucose content determination reagent (RSBIO, Shanghai, China) was balanced at room temperature, the reagent working solution was configured, and 20 ml R1 reagent and 20 ml R2 reagent were mixed well in the same amount. The EP tubes were marked as a blank tube, a calibration tube, a quality control tube and a sample tube accordingly. There are six tubes in every group, each of which is added with 1,000 ul working fluid. Then blank tube was added with 10ul distilled water; calibration tube was added with 10 ul calibration product; quality control tube was added with 10ul quality control product ; sample tube was added with 10 ul samples. After full mixing, it was placed in 37 °C water bath for 15 min. The 200 ul of each tube was transferred to 96-well plate, and the absorbance was measured at the wavelength of 505 nm.

### Detection of HepG2 cell viability by CCK8
The original culture medium was abandoned and the drug was added according to the above administration concentration. The system of each well was 200 μl and the effect was 48 h. The experiment was repeated three times. After drug action, each well was added with 10 μL CCK8 detection solution(UNOCI, Hangzhou, China). The medium was set as a blank control well, and light-avoidance reaction was conducted at 37 °C for 2 h Optical density (OD) was measured at 450 nm and 650 nm. The results were calculated as follows: cell survival rate (%) = OD value of experimental group/OD value of non-drug group ×100%.

### Detection of ROS of HepG2 cells by flow cytometry
The cryopreserved tube contianing 1.5 mL cells was taken out from the liquid nitrogen tank and quickly placed in water bath at 37 °C for about 2 min. The cell suspension in the tube was moved into a 15 mL centrifuge tube, to which five mL complete medium was added, and centrifugal at $300 \times g$ for 5 min at room temperature. After removing the supernatant, cells were resuspended with a moderate amount of complete medium heavy precipitation, inoculated in a 10 cm petri dish, added with complete medium to 10 mL, and cultured again in 37 °C and 5% CO2, saturated humidity. 1 μL ROS probe (Beyotime, Shanghai, China) was added to the resuscitated cells in the proportion of 1:1,000, mixed, incubated at 37 °C for 20 min, and oscillated several times per 5min. After 5-centrifugation, cell

precipitation was collected, one mL PBS was resuscitated, and centrifuged at 500× g for 5 min. one mL PBS was resuspended to be measured. A negative control group was set up and treated in the same way without adding the probe. FITC signal was detected by flow cytometry, FITC signal was detected by FL-1A channel, PI signal was detected by FL-2A channel.

## RNA library construction and sequencing

Total RNA was extracted using Trizol reagent (Invitrogen, CA, USA) following the manufacturer's procedure. The total RNA quantity and purity were analysis of Bioanalyzer 2100 and RNA 6000 Nano LabChip Kit (Agilent, CA, USA) with RIN number >7.0. Approximately 10 ug of total RNA representing a specific adipose type was used to deplete ribosomal RNA according to the manuscript of the Epicentre Ribo-Zero Gold Kit (Illumina, San Diego, USA). Following purification, the poly(A)-or poly(A)+ RNA fractions is fragmented into small pieces using divalent cations under elevated temperature. Then the cleaved RNA fragments were reverse-transcribed to create the final cDNA library in accordance with the protocol for the mRNA-Seq sample preparation kit (Illumina, San Diego, USA), the average insert size for the paired-end libraries was 300 bp (±50 bp). And then we performed the paired-end sequencing on an Illumina Hiseq 4000 at the (LC Bio, China) following the vendor's recommended protocol (150 bp × 2).

### Genome mapping and transcripts assembly

Firstly, Cutadapt (version 1.10) was used to remove the reads that contained adaptor contamination, low quality bases and undetermined bases. Then sequence quality was verified using FastQC (http://www.bioinformatics.babraham.ac.uk/projects/fastqc/). We used Bowtie (version 2) and Tophat (version 2.0) to map reads to the genome (GRCh38)of IR- HepG2 cells. The mapped reads of each sample were assembled using StringTie (version 1.3.0). Then, all transcriptomes from IR- HepG2 cells were merged to reconstruct a comprehensive transcriptome using perl scripts. After the final transcriptome was generated, StringTie and Ballgown [R package] was used to estimate the expression levels of all transcripts.

### LncRNA identification

First of all, transcripts that overlapped with known mRNAs and transcripts shorter than 200 bp were discarded. Then we utilized CPC [0.9-r2], CNCI [2.0] to predict transcripts with coding potential. All transcripts with CPC score >-1 and CNCI score >0 were removed. The remaining transcripts with class code (i, j, o, u, x) were considered as lncRNAs. (i) a transfragfalling entirely within a reference intron(intronic); (j) potentially novel isoform or fragment at least one splice junction is shared with a reference transcript; (o) generic exonic overlap with a reference transcript; (u) unknown, intergenic transcript(intergenic); (x) Exonic overlap with reference on the opposite strand (antisense).

### Target gene prediction, functional and property analysis of differentially expressed mRNAs and lncRNAs

StringTie was used to perform expression level form RNAs and lncRNAs by calculating fragments per kilobase of exon per million fragments mapped (FPKM). The differentially
expressed mRNAs and lncRNAs were selected with |log2(fold change)|>1 and with statistical significance (P value < 0.05) by R package Ballgown To predict the possible functions of the target mRNAs and to explore the pathways in which they participate, target mRNAs were further studied using the GO (http://www.geneontology.org) and KEGG (http://www.genome.jp/kegg) databases. The P value <0.05 and |log2(fold change)|≥1 were defined as statistically significant. The characteristics such as gene lengths, transcript lengths, lncRNA types, exons number and isoforms number of differentially expressed mRNAs and lncRNAs were analyzed and compared according to the reference genome annotation and databases. The ORF lengths of differentially expressed mRNAs and lncRNAs were obtained by genescan (*Burge & Karlin, 1997*)prior to analysis and comparison.

### ceRNA network construction

Using existing miRNA target prediction methods, lncRNA–miRNA–mRNA interactions were identified. TargetScan 5.0 software (http://www.targetscan.org/) and miRanda 3.3a (http://www.microrna.org/microrna/home.do) were simultaneously used to predict the lncRNAs and 3′ UTR sequences of mRNAs as miRNA targets. Based on the ceRNA hypothesis, we also identified putative lncRNA–miRNA integrations using TargetScan and Miranda, these lncRNAs are regarded as competing endogenous RNAs(ceRNA) to bind miRNA competitively and affect mRNA (miRNA target gene) expression indirectly. Combining the two data sources, we got miRNA-lncRNA -mRNA interactions. In order to observe the relationship of the triplet in the IR and identify the APBBR-treatment related lncRNAs. We also mapped 3 IR-associated mRNAs and 7 differential expressed lncRNAs shows opposite tendency before and after APBBR treatment into global triple network. The mutual interaction network of differentially expressed genes and the core-regulatory network of lncRNAs - miRNAs -target genes were mapped via Cytoscape software (V. 3.2.1).

## Statistical analysis

All experimental data were performed using SPSS 23 software (IBM SPSS, Armonk, NY, USA) and R package 3.3.0. All graphical values were presented as the means ± SEM. Student's t test was used to evaluate the statistical differences between two groups. Data among three groups were analyzed with one-way ANOVA followed by LSD when equal variances assumed, and Dunnett's-T3 when equal variances not assumed. A value of $P < 0.05$ was statistically significant different. Figures were drawn by GraphPad Prism 6.02 and Adobe Illustrator CS6.

## RESULTS

### High-dose insulin induced decrease of glucose content in IR-HepG2 cells

The results of intracellular glucose content determination in HepG2 cells are shown in Fig. 1A. Compared to control group, the glucose content in model group decreased significantly ($^{\Delta\Delta}P < 0.01.$)
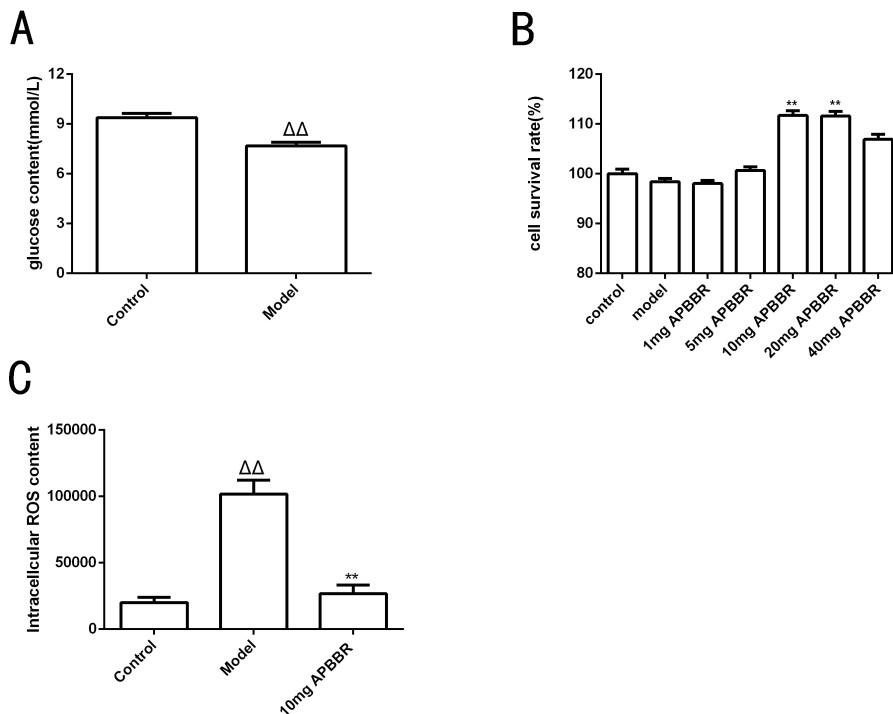

**Figure 1** **Study on the effect of IR-hepG2 cell phenotype.** Glucose concentration of HepG2 cells (A). Survival rate of HepG2 cells (B). The ROS content in HepG2 cells (C). $^{\Delta}P < 0.05$ and $^{\Delta\Delta}P < 0.01$, compared with those in the control group; $*P < 0.05$ and $**P < 0.01$ compared with that of the model group.

## APBBR administration increased the survival rate of IR-HepG2 cells

CCK8 assay was used to detect the proliferation activity of HepG2 cells, as shown in Fig. 1B. The cell survival rate was significantly increased in 10 mg and 20 mg APBBR group compared to model group ($**P < 0.01$). The optimal dosage of APBBR was 10 mg and 20 mg. For convenience, 10 mg was used in the subsequent experiments.

## APBBR decreased ROS level in IR-HepG2 cell model

The ROS content in HepG2 cells detected by flow cytometry were shown in Fig. 1C. $^{\Delta}P < 0.05$ and $^{\Delta\Delta}P < 0.01$, compared with those in the control group; $*P < 0.05$ and $**P < 0.01$ compared with that of the model group. The results showed that the level of ROS in IR-HepG2 cells increased significantly, and APBBR could decrease the level of ROS in IR-HepG2 cell model.

## Expression profiles of mRNAs in experimental groups and ROS related mRNAs

Total RNA and whole transcriptome sequencing data (mRNA, miRNA, lncRNA) were acquired from qualified biospecimens (see ''Methods''). The Illumina Hiseq2500 platform was used to obtain mRNA, lncRNA and miRNA data. As a result of this analysis, we detect thousands of genes. To determine the regulation of mRNA expression, we performed an unsupervised clustering analysis of the significantly regulated genes in IR and APBBR

cells. The EBseq algorithm was applied to filter the differentially expressed genes(DEGs) according to the criteria of a|log2 (fold change)| ≥ 1and a *P*-value <0.05. There were 157 genes whose changes met the criteria. 50 genes were significantly and differentially expressed in Model group relative to Control group, with 24 up-regulated genes and 26 down-regulated genes. 107 genes were significantly altered in APBBR group compared with that in Model group, with 56 genes up-regulated and 51 genes down-regulated Fig. 2A. The Volcano plots for mRNAs are showed as Figs. 2B, 2C. The heatmaps for DEGs are showed as Figs. 2D, 2E. To analyze the function of these DEGs, we next conducted GO Figs. 2F, 2G, and KEGG pathway analysis on these genes Fig. 2H. DEGs in Model vs. Control were largely involved in lysine degradation, RNA degradation, taurine and hypotaurine metabolism, sulfur relay system, ascorbate and aldarate metabolism, AMPK signaling pathway. DEGs in APBBR vs. Model were involved in Protein processing in endoplasmic reticulum, spliceosome, systemic lupus erythematosus, steroid hormone biosynthesis, tryptophan metabolism, MAPK signaling pathway. According to GO and KEGG pathway results as well as previous reports of IR, we notice that genes such as SCD, DUSP5 and PLK3were quite important for ROS development, which may result in IR Fig. 2I.

## Differentially expressed lncRNAs in experimental groups and lncRNAs that participates in APBBR treatment

A total of 66,980 lncRNAs were annotated in this study. In this study, 12,962 lncRNAs were first annotated as known lncRNAs (database: GRCh38 Ensembl v88).The 32,433 newly identified lncRNAs included 17,508 lncRNAs (category code u), 85 antisense lncRNAs, 526 intron lncRNAs, and 3466 other lncRNAs. Almost all known lncRNAs are distributed on each chromosome, but those universal exon lncRNAs and reference lncRNA transcriptors (category code o) do not show significant chromosome location preference Fig. 3A.

From the RNA-seq results, a total of 521 lncRNAs were detected with *P* < 0.05.Compared to Control, 384 lncRNAs were detected to be differentially expressed in Model (89 up regulated and 295 down regulated). Compared to Model, 196 lncRNAs were identified to be differentially expressed in APBBR (113 up regulated and 83 downregulated) Fig. 3B. The Volcano plot for lncRNAs are showed respectively as Figs. 3C, 3D. Heatmap for top 100 differentially expressed lncRNAs in different groups are showed as Figs. 3E, 3F.

If APBBR can improve IR, it is necessary to ensure that the lncRNA expression trend is exactly opposite in both Model vs. Control and APBBR vs. Model. That is to say, we should find out genes decreased in Model group compared to Control group but increase in APBBR group contrasted with Model group (decrease-increase type) and genes increased in Model group compared to Control group but decreased in APBBR group in contrast to Model group (increase-decrease type). According to this principle, Venn analysis was additionally applied to learn the lncRNAs that participates in the exertion of APBBR treatment effect, including increasing/decreasing type and decreasing/increasing type. We found two of them(CTD-2600O9.2, MIR4435-2HG) can be reversed by APBBR in the 295 downregulated genes following IR. Similarly, APBBR treatment tend to reversed five (RP5-1057I20.5, CTD-3014M21.1, HOXB-AS3, LAMA5-AS1, CTD-2517M22.14) of the 89 upregulated lncRNAs following IR (Figs. 3G, 3H).

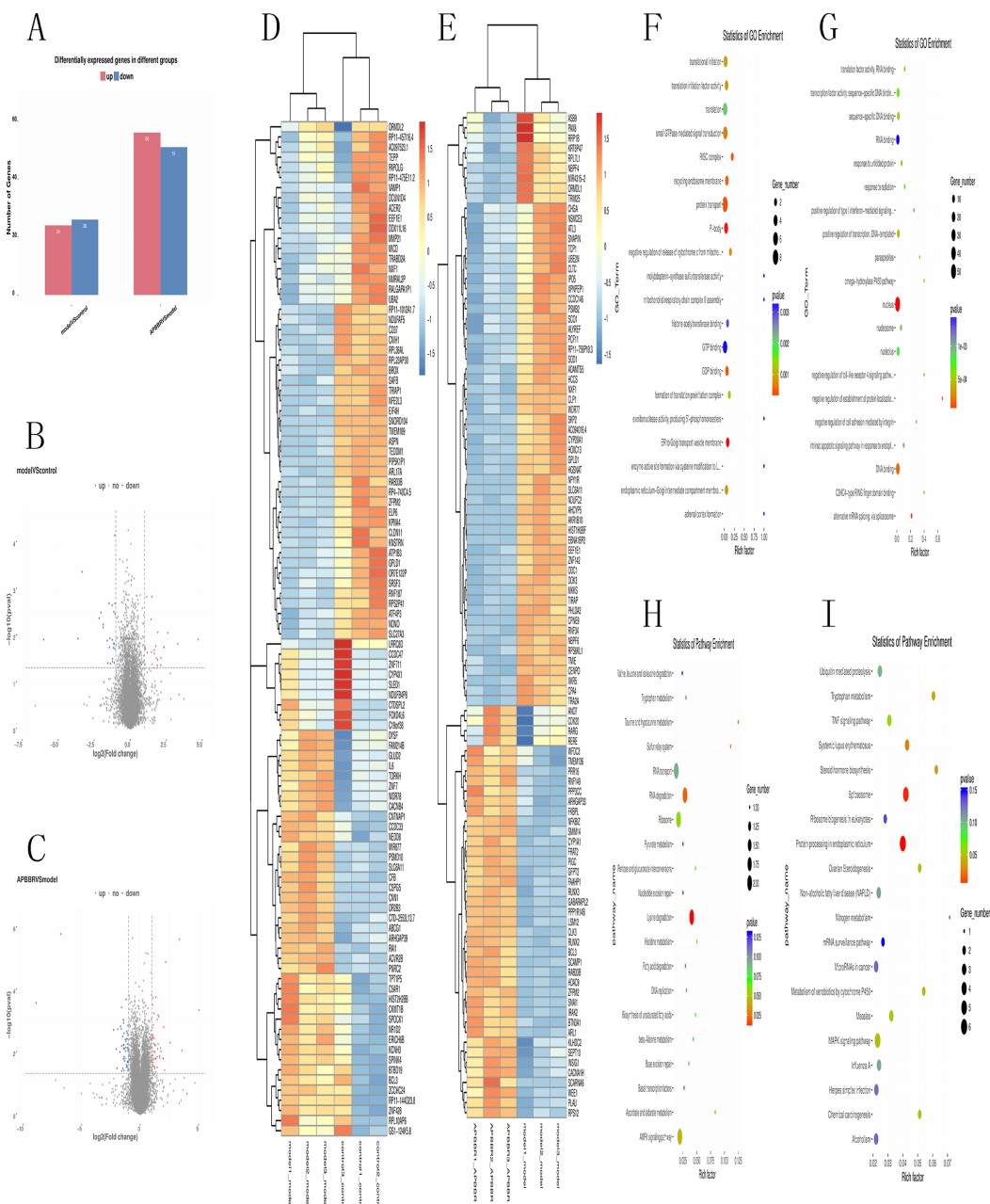

**Figure 2 Expression profiles of mRNAs in experimental groups.** Differential mRNA expression profile (A). Volcano plot depicting DEGs (B, C). Heatmap for DEGs (D, E). GO analysis results for DEGs (F, G). KEGG pathway analysis for DEGs (I).

## Comparative characteristics analysis of mRNAs and lncRNAs

To investigate the difference between differentially expressed mRNAs and lncRNAs in genomic characteristics, transcript lengths, the gene lengths, exons number, ORF lengths, isoforms number were analyzed. FPKM data shows that the abundance of lncRNA is lower than that of mRNA in RNA-seq samples, indicating that lncRNAs exhibited a notable bias

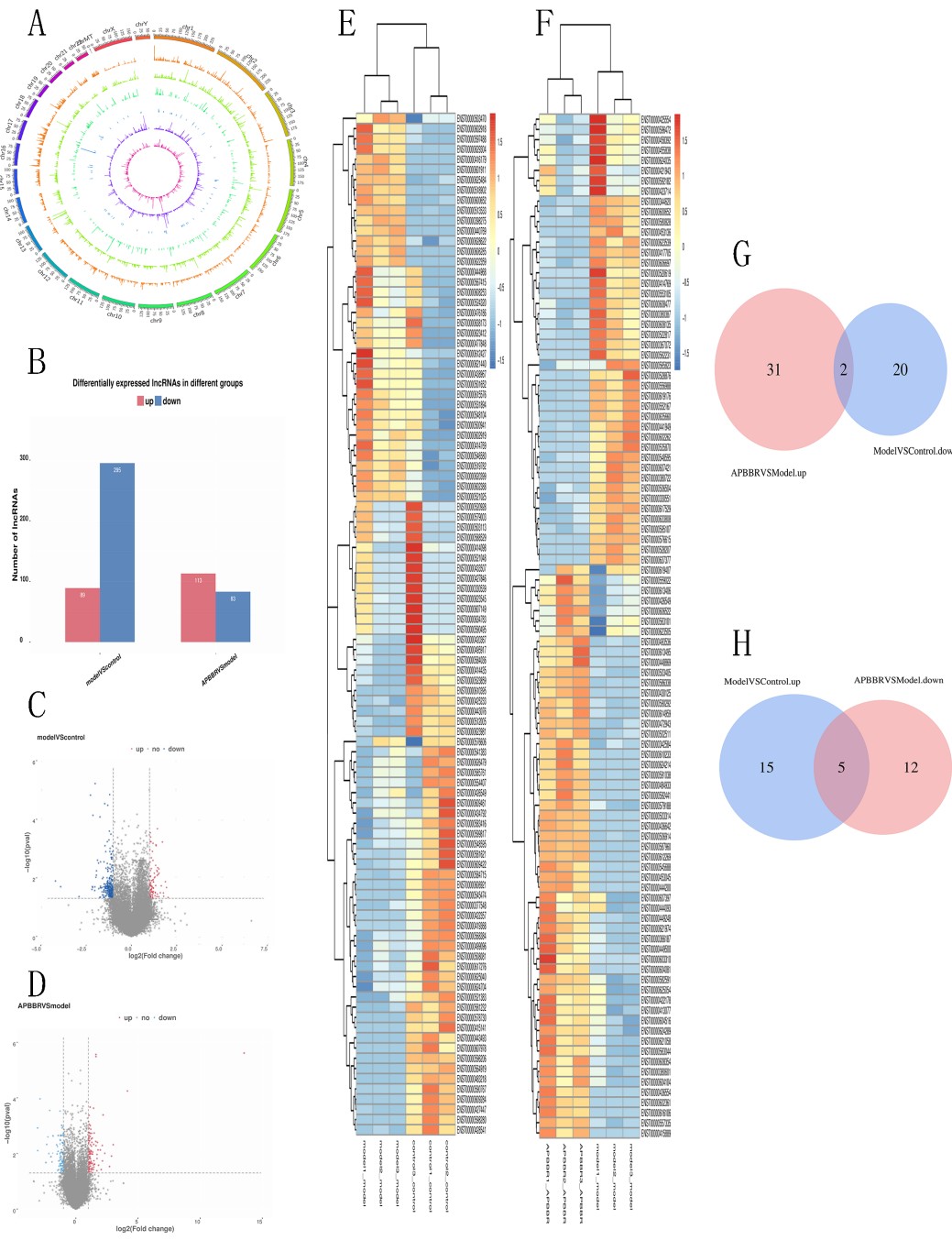

**Figure 3  Differentially expressed lncRNAs in experimental groups.** Circos plots showing all of the lncRNAs in experimental groups (A). Differential lncRNA expression profile (B). Volcano plot depicting lncRNAs (C, D). Heatmap for differentially expressed lncRNAs (E, F). Venn diagrams show overlaps of differentially expressed lncRNAs between experimental groups (G, H). Two lncRNAs decreased in Model group but increased in APBBR group. Five lncRNAs increased expression in the Model group but decreased in the APBBR group.

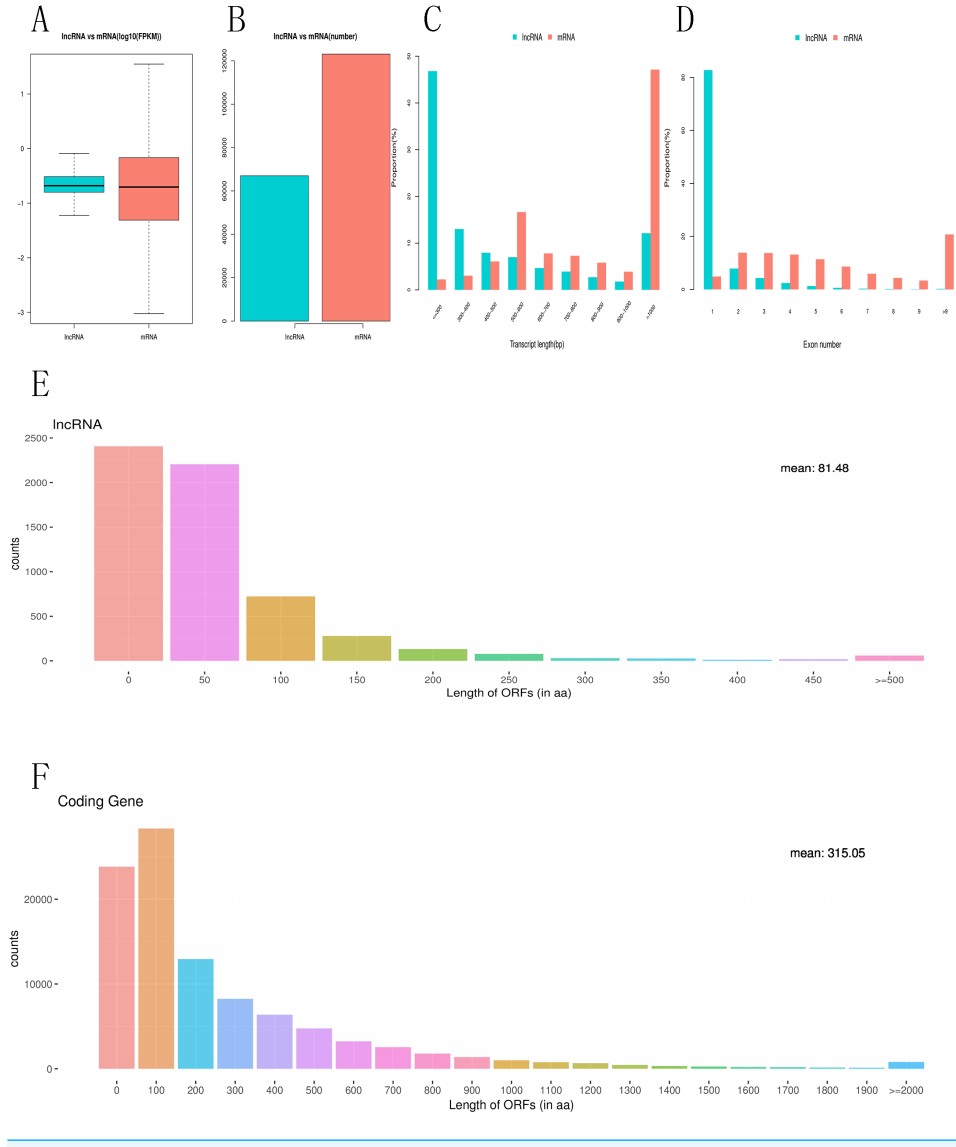

**Figure 4** **Comparative characteristics analysis of mRNAs and lncRNAs.** Expression level of mRNA and lncRNA (A, B). length distribution of mRNA and lncRNA (C). Exon number of mRNA and lncRNA(D). ORF length distribution of mRNA and lncRNA (E, F).

toward fewer transcript isoforms Figs. 4A, 4B. In addition, we found that the length of lncRNA is usually shorter than that of mRNA Fig. 4C. For example, most lncRNAs have less than six exons, while mRNA have more exons and exon numbers distributed over a wider range. Some mRNA have as many as 30 exons Fig. 4D. LncRNAs also have shorter (60–90 nucleotide) ORF than mRNA, while most mRNAs have more than 500 nucleotide ORF Figs. 4E, 4F.

## Construction of a lncRNAs–miRNA–mRNA network

To systematically explore the influence of dynamic changes in ceRNA regulation on gene expression related to IR and ROS in Normal, Model and APBBR-treated hepG2 cells.

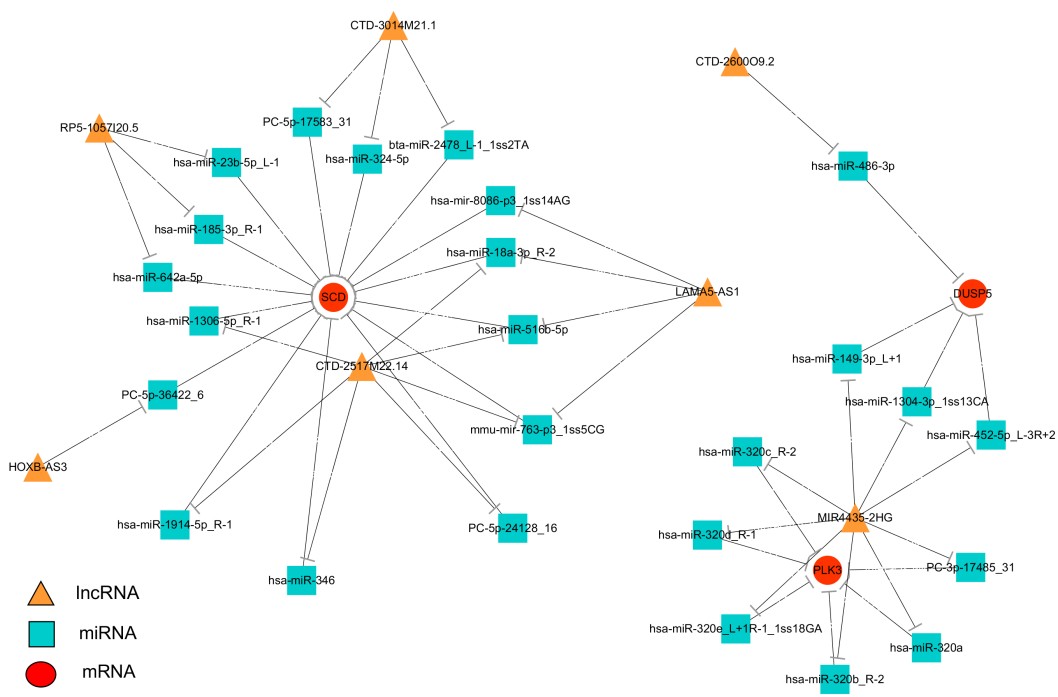

**Figure 5** **Interaction network of differentiated lncRNA–miRNA–mRNA in normal, IR and APBBR–treated HepG2 cells.** Rectangle, triangle and ellipse represent miRNA, lncRNA and mRNA, respectively. This genes figure are showed genes belonging to the same trend (increase-decrease type/decrease-increase type). In the lncRNA–miRNA–mRNA predicted interaction network, 30 miRNAs are predicted to be possible target genes for regulating them.

we constructed a ceRNA network, integrating matched expression profiles of lncRNAs, miRNAs and mRNAs. Target genes were selected according to the results of GO or KEGG pathway analysis or according to previous reports of ROS-associated genes that may lead to IR, and then the global miRNA-mediated ceRNA network was constructed (Fig. 5). In the lncRNA-miRNA-mRNA predicted interaction network, seven lncRNAs and three mRNAs had the same expression trend (belonging to the increase-decrease type/decrease-increase type). Twenty-five miRNAs were predicted to be possible target genes for regulation.

## DISCUSSION

Systemic or local IR occurs when insulin target organs or tissues become less sensitive and responsive to endogenous or exogenous insulin (*Semenkovich, 2016*). IR refers to the weakened response of tissues to circulating insulin (*Archer, Von Schulze & Geiger, 2018*), it mainly take place in skeletal muscle cells, fat cells and liver cells, it can also occur in vascular endothelium and islet beta cells (*Dontsov & Vasil'eva, 2016*). IR involving multiple molecules and signaling mechanism, including insulin and its antagonist, insulin receptor substrates, phosphatidyl inositol 3 kinase (PI3K), glucose transporter gene and protein, mitogen-activated protein kinase (MAPK), etc (*Seino et al., 2010*). It has been proved that IR is the common pathological basis of abnormal glucose and lipid metabolism of essential

hypertension, T2DM,coronary heart disease, hyperlipidemia and other diseases (*Tan, Sasagawa & Mori, 2017*; *Otto et al., 2019*).

ROS include superoxide ionon ($O^{2-}$), hydroxyl radical ($OH^{-}$) and hydrogen peroxide ($H_2O_2$), among which $O^{2-}$ has high activity and cytotoxicity and is mainly produced by aerobic microorganisms (*Newsholme et al., 2016*). If the cell antioxidant system fails to suppress ROS, ROS would react with cell macromolecules, leading to lipid peroxidation, causing cell DNA damage, which may result in oxidative stress and oxidative damage to the cell (*Chattopadhyay et al., 2015*). Oxidative stress is also known as ROS imbalance, where reactive oxygen species produced by the body exceeds their antioxidant capacity, leading to excessive generation of free ROS or dysfunction of antioxidant system (*Meex, Blaak & Van Loon, 2019*). Oxidative stress can damage critical cellular macromolecules and stimulate various stress sensitive intracellular pathways such as c-Jun, N-terminal kinase, ERK1/2 and the NF-$\kappa$B (*Keane et al., 2015*). It can also produce chronic low-grade inflammatory reaction (*Klaunig et al., 2011*). It is increasingly accepted that oxidative stress is a significant contributor to the progression of IR.

It is noteworthy that IR also produces excessive free radicals and superoxides through proton electrochemical gradient, which destroys the antioxidant defense ability of various tissues (*Valko et al., 2007*), and also affects the expression of glucose transporter-GLUT1 and antioxidant defense function, thus increasing oxidative stress (*Gonzalez-Menendez et al., 2018*).

Astragalus Polysaccharide (AP) and berberine (BBR) have been reported to have antidiabetic properties (*Meng et al., 2018*; *Jiang et al., 2015*). AP can modulate the insulin-initiated phosphorylation cascades in a similar manner as metformin in HepG2/IR cell model (*Sun et al., 2019*). BBR is reported to demonstrate an excellent activity with oxidative stress and inflammation through a variety of signaling pathways including NF-$\kappa$B, AMPK, Nrf2/HO, JNK, c-Jun and MAPKs pathways and various kinases in cell (*Ma et al., 2018*) which indicates its IR-improving potential. However, the molecular mechanisms for this action are not fully clarified. Our study showed that APBBR can improve the proliferation activity and decrease the level of ROS in IR-HepG2 cell model, suggesting the antioxidant mechanism of APBBR and APBBR may be a stable and reliable means to attenuate IR and its adverse metabolic consequences in HepG2 cells.

RNA-seq analysis is a powerful approach for investigating underlying molecular mechanisms of complex biological processes (*Han et al., 2014*). NcRNAs include miRNAs and LncRNAs. LncRNA is a hotspot in the field of biological medicine, its important function in the cell has been widely accepted as the successor of another wave of research followed miRNAs. It was found that the expression of lncRNA in human pancreatic island cells had exceeded 1,100 species (*Moran et al., 2012*). Various miRNAs and lncRNAs have been found to be associated with IR and intensive studies on ncRNAs had shed light on understanding the pathogenesis of IR (*Esguerra et al., 2011*; *Davalos et al., 2011*; *Zeggini et al., 2007*). Recently, competing endogenous RNA networks have been reported as a novel mechanism to set up an extensive regulatory network in a post-transcriptional layer. According to the "ceRNA hypothesis", lncRNAs competitively interacted with the miRNAs and thereby "talk" to mRNAs through microRNA response elements (MREs) (*Phelps et al.,*

*2016*). Based on this reasoning, we constructed a global triple network ,in which lncRNA and mRNA from a triple sharing the same miRNA.

To the best of our knowledge, this was the first study to detect genome-wide transcriptome of IR-HepG2 cell to predict potential lncRNA-mediated networks in APBBR attenuating IR at the molecular level. In the present study, we found that compared with Control group, 384 DElncRNAs and 50 DEGs were present in Model group and there were 196 DElncRNAs and 107 DEGs in APBBR vs. Model. GO and KEGG pathway analyses showed the potential functions of the DEGs involved in the mechanism of APBBR treatment; and the ceRNA analysis identified several potential lncRNA/miRNA/mRNA interaction networks that might ameliorate the development of IR. The sequencing depth involves mRNA, miRNA and lncRNA, which allowed us to detect differences in gene expression at the transcriptome level and predict the regulatory mechanisms that may mediate ROS changes by ceRNA. These results suggest more comprehensive perspectives to understand IR progress and explore mechanisms of APBBR treatment. Firstly, we select the candidate mRNA according to GO and KEGG pathway analyses and previous relevant studies of ROS and IR. Secondly, for validation, we chose the lncRNAs that exhibited opposite variation trend before and after APBBR treatment. Venn analysis was additionally applied to learn the possible marker that participates in the process of APBBR's improving IR.

The ceRNA network offers considerable clues for understanding the key roles of ceRNA-mediated gene regulatory networks in ROS genesis and APBBR regulation mechanism as well as further detail analysis of biological function of lncRNA. From the ceRNA network, we identified a substantial amount of cross-talk within the non-coding RNAs, in which mRNAs were co-expressed with candidate lncRNAs mediated by miRNAs and subsequently formed a complex network. According to this analysis, we discovered 7 lncRNAs involving in the process of APBBR's improving IR. Of which, the lncRNA MIR4435-2HG decreased expression in Model group but increased in APBBR group, connecting many mRNAs that may ameliorate IR according to GO enrichment analysis and pathway analysis, and it could directly interact with aberrant miRNA expression contributes to IR. For instance, miR-320 is reported to promote IR in high glucose treated adipocytes and anti-miR-320 oligo was found to regulate IR in adipocytes by improving insulin–PI3-K signalling pathways (*Ling et al., 2009*). In our study, miR-320 increased expression in Model group but decreased after APBBR treatment. According to the ceRNA theory, there may be a triplet that contained MIR4435-2HG, miR-320 and the downstream target mRNAs, the lncRNA could sponge the miRNA and change the expression of target mRNAs. We found that PLK3 and DUSP5 tend to be the target mRNAs in the network. Polo-like kinase 3 (Plk3) is a serine/threonine protein kinase of the Polo-like kinase family, which is critical for tumor suppression and stress responses (*Barr, Sillje & Nigg, 2004*). Moreover, up-regulation of Dusp5 (a tumor suppressor) is able to negatively regulate the MAPK signaling pathway (*Zhang et al., 2014*). MAPK plays a critical role during the process of inflammation and IR (*Zhou et al., 2018*), the negative regulation of MAPK might ameliorate inflammation and activate insulin signaling in liver. That is to say, miR-320 may cause IR by activating the MAPK pathway

to increase hepatic oxidative stress and APBBR may decrease ROS level in IR-HepG2 cells by negatively regulating miR-320.

IR can play a role in lipid metabolism disorders in the body and promotes lipid deposition in liver. We notice that the mRNA related to lipid metabolism SCD increase significantly in IR vs. Control and it tend to be decreased in APBBR vs. IR. SCD encodes an enzyme involved in fatty acid biosynthesis, primarily the synthesis of oleic acid. The enzyme is also important in the conversion of polyunsaturated fatty acids (PUFAS) to monounsaturated fatty acids (MUFAS). The PUFAs are the major cellular target of ROS and readily provide an extractable proton to oxygen-free radicals (*Ushio-Fukai & Alexander, 2004*). The decreased expression of SCD after APBBR treatment may decrease ROS level in IR-HepG2 cells and thereby attenuating IR. These results are consistent with previous studies that SCD has been shown to be over expressed in fat tissues with obesity and metabolic disorders and absence of SCD can improve metabolic syndrome in mice (*MacDonald et al., 2008*). Using ceRNA, we narrowed the targets of mRNAs related to IR and ROS with a lncRNA_miRNA_mRNA network, pointing the way for further studies to verify the genes and describe their function.

Generally, we believe that biochemical experiments combined with high-throughput technologies could shed new light to explore complex biological mechanisms underlying the efficacy of APBBR. At the same time, the lncRNA played an important role in the IR and could offer new therapeutic targets for IR mechanism research. It is noteworthy that many lncRNAs in the ceRNA network are not annotated, which is quite worth futher study. We hope to inspire researchers to study the role of non-coding RNAs in IR and APBBR treatment.

## CONCLUSIONS

In summary, our investigation indicates that attenuated IR in hepG2 cells manifestation by increased cell survival rate and decreased ROS level. Furthermore, we explored the target genes highly enriched in pathways related to ROS and IR and predicted the network regulated by lncRNA according to ceRNA theory, revealing the potential biofunctional roles for lncRNAs as drivers of IR. Transcriptome profiling and regulatory network provides a rationale for exploiting the insulin-sensitizing potential of APBBR by decreasing ROS level via MAPK pathway, which enriches the therapeutic options in the treatment of IR and enhanced our understanding of APBBR. This study provided the expression profile of lncRNAs in IR-hepG2 cells for the first time and laid a foundation for the study on the molecular mechanism of lncRNA regulating IR as well as the follow-up study on the lncRNA regulating mechanism of APBBR in treating IR.

### Funding

This study was supported by the grants from the National Natural Science Foundation of China (No. 81603351). The funders had no role in study design, data collection and analysis, decision to publish, or preparation of the manuscript.

## Grant Disclosures

The following grant information was disclosed by the authors:
National Natural Science Foundation of China: No. 81603351.

## Competing Interests

Zhu-Jun Mao is a co-inventor of patent no. CN108478591A.

## Author Contributions

- Min Lin performed the experiments, analyzed the data, prepared figures and/or tables, authored or reviewed drafts of the paper, and approved the final draft.
- Zhu-Jun Mao conceived and designed the experiments, authored or reviewed drafts of the paper, and approved the final draft.

## Patent Disclosures

The following patent dependencies were disclosed by the authors:
Zhu-Jun Mao is a co-inventor of patent no. CN108478591A.

## Data Availability

The raw measurements are available in the Supplementary Files.

## Supplemental Information

Supplemental information for this article can be found online at http://dx.doi.org/10.7717/peerj.8604#supplemental-information.

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
