# Peer review of "lncRNA–mRNA competing endogenous RNA network in IR-hepG2 cells ameliorated by APBBR decreasing ROS levels: a systematic analysis"

_PeerJ, doi:10.7717/peerj.8604_

## Round 0.1 · original submission · Major Revisions

Although all the reviewers acknowledge the originality of the study, several issues have been raised by the reviewers in different aspects of the work that need to be addressed (basic reporting, experimental design, validity of the findings). English should be extensively improved.

Reviewer 1 ·

Basic reporting

English is not clear at all. There is no continuity between sentences. Very few literatures were cited. Poor description of aim.

Experimental design

Poorly designed experiments. Methods not fully described. Result section poorly written not clear.

Validity of the findings

no comments

Additional comments

1. English is poorly written , not clear what authors want to say.
2. Author should describe aim in detail.
3. Figures are not properly labelled. Figures are very confusing and author should increase the resolution of figures
4. Introduction is very very small. Author should describe about ROS and what are the relationship between ROS and Diabetes?
5. why authors chose only HepG-2 cell lines, why not Huh7 cell lines?
6. Did the authors looked into total ROS? Phagolysosomal ROS? or Mitochondrial ROS?
7. Figure 1c is POORLY represented. what is the meaning of triangles and stars? what does model represent?
8. line 120 what is grpm?
9. why did the authors chose to add 10mg APBBR? Did the authors perform standard curve?
10. Author should take care while writing P value .

Reviewer 2 ·

Basic reporting

The study is overall well presented, though with several grammar mistakes and confusing statement. Introduction provided sufficient background information about the study. The structure of the manuscript is fine. This article is self-contained with relevant results to hypotheses.

Experimental design

HepG2 IR cell model was used to generate the model. Then unbiased genome-wide RNA sequencing was used to investigate changes between groups. With the results, GO and
KEGG were then adopted to interpret the change. Venn analysis screened out candidate lncRNAs and target genes followed by construct interaction network of differentiated
lncRNA–miRNA–mRNA. The design is overall logic, with description sufficiet.

Validity of the findings

The idea of the study is good, with novelty. But was hust a preliminary exploration, still needs further molecular validation.

Additional comments

This article is overall well-designed which brings up an interesting question in discussing combined activities of the two potential antidiabetic ingredients, AP and BBR in ameliorating IR and efficacy in treating diabetes. But there're still some issues need to be explained as follows.
1. Validation reaults are needed to support the solidity of the study.
2. Interaction network of differentiated lncRNA–miRNA–mRNA needs further interpretation.
3. Sellection critiria of candidate lncRNAs and these target genes needs further concrete explaination.

·

Basic reporting

1. Manuscript has typographical errors in multiple places which need thorough checking. The letters”ff” is shown as box in all places. Kindly verify the font settings. Italicize in vitro in line 370.
2. Language and sentence construction needs refining to make sure the intended meaning is conveyed. Eg: Line 366 states “microorganisms react with cell macromolecules” but it should be “ROS reacts with..”
3. Titles of sub headings in results section should be revised to indicate the results and not just the technique used. The first two sub headings of results section mentions just the technique used. The result obtained is not actually explained even in the first paragraph.
4. Raw data submitted is not proper. (kindly refer general comments for detailed comments).

Experimental design

1. The research is original and has the potential to add significant knowledge upon further refinement.
2. The aim of the manuscript is well framed but the experimental plan has introduced a bias in the selection of genes. This makes the conclusion biased too which need to be clarified by correcting the title of the manuscript, aim and conclusion accordingly. (kindly refer general comments for detailed comments).

Validity of the findings

1. Since the manuscript achieves identifying the target genes through pathway analysis and has not validated them, the discussion should clarify that it is in silico finding and in vitro/ in vivo validation is further needed when discussing about the pathway in figure-5.
2. Previous reports with AP and BBR individually have reported to have anti tumor activity inducing cell death in multiple cancer systems. Therefore it is imperative to test their effect individually and in combination on proliferation/ cell survival of HepG2 control cells as well as model cells to see whether the increase in proliferation seen by the authors is specific to model cells.
3. The glucose content is tested in only control and model cells and not with APBBR treatment. Also change in ROS in control cells upon APBBR is not tested. Adding these data will be more meaningful support the effect of the combination of these drugs in the concentration used in this study.

Additional comments

In the manuscript titled “Systematical analysis of lncRNA–mRNA competing endogenous RNA network on IR-hepG2 cells ameliorated by APBBR through decreasing ROS level”, the authors have attempted to understand the mehcanism of APBBR in regulaitng lncRNAs involved in regulating ROS pathway. Though the manuscript reveals the key lncRNAs and the ceRNA pathways regulated by APBBR to regulate ROS pathway in IR condition by pathway analysis, certain key points need to be addressed.

1. Figure-4 represents comparative analysis between mRNA and lncRNAs does not add any significant information to the aim of this paper. Instead, comparison of mRNA and lncRNAs between control and model with and without APBBR treatment will add up information on the relationship between the ROS and RNA biogenesis in this context and the effect of APBBR if it alters.
2. Line-344 states “24 miRNAs, 7lncRNAs and 3 mRNAs”, but line 741 states “30 miRNAs; Figure-5 shows 6 lncRNAs. Kindly verify these statements.
3. In line-465, Is it “SOD” or “SCD” ?
4. Line-120 states 100grpm. Centrifugation is performed according to xg or rpm calculation. Kindly clarify.
5. Lines 340, 424 states candidates are selected based on relevance to ROS and IR. Therefore, there is a bias to see only the effect of APBBR on the relationship between ROS change in IR. Therefore the aim has to corrected accordingly. Moreover, how does this justify the statement in lines 336-337. Further, the analysis cannot be global pathway analysis in figure-5 when ROS pathway is selectively enriched.
6. The lncRNAs shortlisted from venn diagram, (total of 7) are not seen in the lncRNA differential expression data given in supplementary file.
7. The attached supplementary files does not represent the total list of lncRNAs obtained from analysis because it has only 100 lncRNAS in both the conditions and among them most of them does not have a significant P value. Only 1 in model vs control and 3 in APBBR vs model are significant (P < 0.05).

---

## Round 0.2 · Minor Revisions

Please address both comments raised by Reviewer 3.

Reviewer 1 ·

Basic reporting

English is clear and Professional.

Experimental design

Primary research within the aims and scope of the journal. Research question well defined.

Validity of the findings

Conclusions are well stated and all data are provided

Additional comments

Thank you authors for revising the manuscript.

·

Basic reporting

Work is well presented.

Experimental design

The research is original and has the potential to add significant knowledge to the field.

Validity of the findings

The manuscript identifies the target genes through pathway analysis and can lead to future studies on validation..

Additional comments

1. Kindly look at the lncRNA expression profile file updated as supplementary data and revisit the analysis. Presence of same lncRNA multiple times is acceptable but same lncRNA (SNHG6) is up regulated as well as down regulated (both significantly too). Kindly clarify the analysis method and make sure these anomalies are rectified.

2. As noted earlier, previous reports with AP and BBR individually have reported to have anti-proliferative activity inducing cell death in multiple cell lines including HepG2 cells. But here combination of both increases the proliferation inn HepG2 derived model cells. Therefore it is imperative to test their effect individually and in combination on proliferation/ cell survival of HepG2 control cells as well as model cells to see whether the increase in proliferation seen by the authors in this study is specific to model cells. This data is essential in two aspects. 1. whether AP and BBR have different functions when administered individually from being together. 2. whether the function of AP and BBR changes with context (control cell line and model cell line). So, doing this experiment is critical to this manuscript also.

---

## Round 0.3 · accepted · Accept

The authors have satisfactorily addressed the last issues raised by the reviewer.

·

Basic reporting

Work is well presented with references.

Experimental design

The research is original with methods explained in detail and has the potential to add significant knowledge to the field.

Validity of the findings

The manuscript identifies the target genes through pathway analysis and can lead to future studies on validation of the target genes.